# *Pseudomonas aeruginosa* Bloodstream Infections in Patients with Cancer: Differences between Patients with Hematological Malignancies and Solid Tumors

**DOI:** 10.3390/pathogens11101132

**Published:** 2022-09-30

**Authors:** Cristina Royo-Cebrecos, Julia Laporte-Amargós, Marta Peña, Isabel Ruiz-Camps, Pedro Puerta-Alcalde, Edson Abdala, Chiara Oltolini, Murat Akova, Miguel Montejo, Malgorzata Mikulska, Pilar Martín-Dávila, Fabian Herrera, Oriol Gasch, Lubos Drgona, Hugo Manuel Paz Morales, Anne-Sophie Brunel, Estefanía García, Burcu Isler, Winfried V. Kern, Zaira R. Palacios-Baena, Guillermo Maestro de la Calle, Maria Milagro Montero, Souha S. Kanj, Oguz R. Sipahi, Sebnem Calik, Ignacio Márquez-Gómez, Jorge I. Marin, Marisa Z. R. Gomes, Philipp Hemmatti, Rafael Araos, Maddalena Peghin, José Luis del Pozo, Lucrecia Yáñez, Robert Tilley, Adriana Manzur, Andrés Novo, Jordi Carratalà, Carlota Gudiol

**Affiliations:** 1Internal Medicine Department, Hospital Nostra Senyora de Meritxell, Andorra Health Services (SAAS), AD700 Escaldes-Engordany, Andorra; 2Infectious Diseases Department, Bellvitge University Hospital, IDIBELL, University of Barcelona, 08907 Barcelona, Spain; 3Institut Català d’Oncologia (ICO), Hospital Duran i Reynals, IDIBELL, 08907 Barcelona, Spain; 4Hematology Department, Institut Català d’Oncologia (ICO)–Hospital Duran i Reynals, IDIBELL, 08907 Barcelona, Spain; 5Infectious Diseases Department, Vall d’Hebron University Hospital, 08035 Barcelona, Spain; 6Infectious Diseases Department, Hospital Clínic i Provincial, 08035 Barcelona, Spain; 7Instituto do Câncer do Estado de São Paulo, Faculty of Medicine, Univesity of São Paulo, Sao Paulo 01246, Brazil; 8Unit of Infectious and Tropical Diseases, IRCCS San Raffaele Scientific Institute, 20132 Milan, Italy; 9Department of Infectious Diseases, Hacettepe University School of Medicine, 06230 Ankara, Turkey; 10Infectious Diseases Unit, Cruces University Hospital, 48903 Bilbao, Spain; 11Division of Infectious Diseases, University of Genoa (DISSAL) and Ospedale Policlinico San Martino, 16132 Genoa, Italy; 12Infectious Diseases Department, Ramon y Cajal Hospital, 28034 Madrid, Spain; 13Infectious Diseases Section, Department of Medicine, Centro de Educación Médica e Investigaciones Clínicas (CEMIC), Buenos Aires C1430EFA, Argentina; 14Infectious Diseases Department, Parc Taulí University Hospital, 08208 Sabadell, Spain; 15Oncohematology Department, Comenius University and National Cancer Institute, 81499 Bratislava, Slovakia; 16Infectious Diseases Department, Hospital Erasto Gaertner, Curitiba 81520-060, Brazil; 17Infectious Diseases Department, Department of Medicine, Lausanne University Hospital, (CHUV), 1011 Lausanne, Switzerland; 18Hematology Department, Reina Sofía University Hospital-IMIBIC-UCO, Córdoba 14004, Argentina; 19Department of Infectious Diseases and Clinical Microbiology, Istanbul Education and Research Hospital, 34668 Istanbul, Turkey; 20Division of Infectious Diseases, Department of Medicine II, University of Freiburg Medical Center and Faculty of Medicine, 79106 Freiburg, Germany; 21Unit of Infectious Diseases and Clinical Microbiology, Virgen Macarena University Hospital, Institute of Biomedicine of Seville (IBIS), 41013 Seville, Spain; 22Infectious Diseases Unit, Instituto de Investigación Hospital “12 de Octubre” (i+12), “12 de Octubre”, University Hospital, School of Medicine, Universidad Complutense, 28041 Madrid, Spain; 23Infectious Diseases Service, Hospital del Mar, Infectious Pathology and Antimicrobials Research Group (IPAR), Institut Hospital del Mar d’Investigations Mèdiques (IMIM), Universitat Autònoma de Barcelona (UAB), CEXS-Universitat Pompeu Fabra, 08003 Barcelona, Spain; 24Infectious Diseases Division, American University of Beirut Medical Center, Beirut 1107 2020, Lebanon; 25Faculty of Medicine, Ege University, 35040 Izmir, Turkey; 26University of Health Science Izmir Bozyaka Training and Research Hospital, 35170 Izmir, Turkey; 27Infectious Diseases Department, Hospital Regional de Málaga, 29010 Málaga, Spain; 28Infectious Diseases and Clinical Microbiology Department, Clínica Maraya, Pereira, Colombia. Critical Care and Clinical Microbiology Department, Manizales 170001-17, Colombia; 29Hospital Federal dos Servidores do Estado, and Instituto Oswaldo Cruz, Fundação Oswaldo Cruz, Ministério da Saúde, Rio de Janeiro 20221-161, Brazil; 30Department of Hematology, Oncology and Palliative Care, Klinikum Ernst von Bergmann, Academic Teaching Hospital, Charité University Medical School, 10117 Berlin, Germany; 31Instituto de Ciencias e Innovación en Medicina, Facultad de Medicina Clínica Alemana Universidad del Desarrollo, Santiago de Chile 12461, Chile, and Millennium Initiative for Collaborative Research on Bacterial Resistance (MICROB-R);; 32Infectious Diseases Clinic, Department of Medicine, University of Udine and Azienda Sanitaria Universitaria Integrata in Udine, and Infectious and Tropical Diseases Unit, Department of Medicine and Surgery, University of Insubria-ASST-Sette Laghi, 33100 Udine, Italy; 33Infectious Diseases and Microbiology Unit, Navarra University Clinic, 31008 Pamplona, Spain; 34Hematology Department, Marqués de Valdecilla University Hospital, 39008 Santander, Spain; 35Microbiology Department, University Hospitals Plymouth NHS Trust, Plymouth PL6 8DH, UK; 36Infectious Diseases, Hospital Rawson, San Juan J5400, Argentina; 37Hematology Department, Son Espases University Hospital, 07120 Palma de Mallorca, Spain; 38University of Barcelona, 08007 Barcelona, Spain; 39Centro de Investigación Biomédica en Red de Enfermedades Infecciosas (CIBERINFEC), Instituto de Salud Carlos III, 28029 Madrid, Spain

**Keywords:** *Pseudomonas aeruginosa*, bacteremia, bloodstream infection, cancer, solid tumor, hematologic malignancy

## Abstract

Objectives: To assess the clinical features and outcomes of *Pseudomonas aeruginosa* bloodstream infection (PA BSI) in neutropenic patients with hematological malignancies (HM) and with solid tumors (ST), and identify the risk factors for 30-day mortality. Methods: We performed a large multicenter, retrospective cohort study including onco-hematological neutropenic patients with PA BSI conducted across 34 centers in 12 countries (January 2006–May 2018). Episodes occurring in hematologic patients were compared to those developing in patients with ST. Risk factors associated with 30-day mortality were investigated in both groups. Results: Of 1217 episodes of PA BSI, 917 occurred in patients with HM and 300 in patients with ST. Hematological patients had more commonly profound neutropenia (0.1 × 10^9^ cells/mm) (67% vs. 44.6%; *p* < 0.001), and a high risk Multinational Association for Supportive Care in Cancer (MASCC) index score (32.2% vs. 26.7%; *p* = 0.05). Catheter-infection (10.7% vs. 4.7%; *p* = 0.001), mucositis (2.4% vs. 0.7%; *p* = 0.042), and perianal infection (3.6% vs. 0.3%; *p* = 0.001) predominated as BSI sources in the hematological patients, whereas pneumonia (22.9% vs. 33.7%; *p* < 0.001) and other abdominal sites (2.8% vs. 6.3%; *p* = 0.006) were more common in patients with ST. Hematological patients had more frequent BSI due to multidrug-resistant *P. aeruginosa* (MDRPA) (23.2% vs. 7.7%; *p* < 0.001), and were more likely to receive inadequate initial antibiotic therapy (IEAT) (20.1% vs. 12%; *p* < 0.001). Patients with ST presented more frequently with septic shock (45.8% vs. 30%; *p* < 0.001), and presented worse outcomes, with increased 7-day (38% vs. 24.2%; *p* < 0.001) and 30-day (49% vs. 37.3%; *p* < 0.001) case-fatality rates. Risk factors for 30-day mortality in hematologic patients were high risk MASCC index score, IEAT, pneumonia, infection due to MDRPA, and septic shock. Risk factors for 30-day mortality in patients with ST were high risk MASCC index score, IEAT, persistent BSI, and septic shock. Therapy with granulocyte colony-stimulating factor was associated with survival in both groups. Conclusions: The clinical features and outcomes of PA BSI in neutropenic cancer patients showed some differences depending on the underlying malignancy. Considering these differences and the risk factors for mortality may be useful to optimize their therapeutic management. Among the risk factors associated with overall mortality, IEAT and the administration of granulocyte colony-stimulating factor were the only modifiable variables.

## 1. Introduction

Bloodstream infection (BSI) is a major cause of morbidity and mortality in neutropenic cancer patients. A shift in the etiology of BSI to Gram-negative bacilli (GNB) as well as an increase in antibiotic resistance among them have been reported in cancer patients in the last decades [1,2,3,4]. *Pseudomonas aeruginosa* (PA) has historically been one of the major causes of severe sepsis and high mortality in cancer patients with neutropenia [5,6,7,8]. Thus, the emergence of antimicrobial resistance in PA is of special concern, since initial inadequate empirical antibiotic therapy (IEAT) is associated with increased mortality in this setting [9,10,11,12].

We recently published the results of a large international cohort study (the IRONIC study) in which we found a significant increase in BSI due to multidrug-resistant PA (MDRPA) in neutropenic cancer patients throughout the study period (2006–2018) [13]. In this study, we provided a predictive model for multidrug resistance that can be easily calculated using a web-based calculator (Risk of Multidrug resistance Pseudomonas aeruginosa Bloodstream Infection (MDR-PA BSI). Available online: http://ubidi.shinyapps.io/ironic, accessed on 20 July 2022) and allows for the identification of the patients who may benefit from the early administration of broad-spectrum antibiotic coverage against MDR strains according to the local susceptibility patterns. The cohort of patients consisted of patients with hematologic malignancies (HM) including hematopoietic stem cell transplant (HSCT) recipients, along with patients with solid tumors (ST). Patients with HM and with ST are often considered as a single group of patients when it comes to providing recommendations on the management of febrile neutropenia [14,15]. Nevertheless, these two groups of patients present significant differences in the etiology, clinical presentation, and outcomes of BSI during neutropenia [16]. In fact, the risk factors associated with mortality in patients with BSI also appear to be different, depending on the underlying malignant condition [17].

The available literature on *Pseudomonas aeruginosa* bloodstream infection (PA BSI) usually focuses on hematologic patients [6,7,8,11,18], and there is no available information specifically on patients with ST, who indeed, appeared to be more susceptible to PA BSI than the hematologic patients in the above-mentioned study [13]. The knowledge of the potential differences between patients with HM and ST presenting with PA BSI could be useful in order to improve their management during febrile neutropenia. Thus, the aim of this study was to identify the differences in the clinical presentation, rates of multidrug resistance, source of infection, outcomes, and risk factors for the 30-day mortality of BSI due to PA in patients with HM compared to patients with ST, analyzing the large cohort of neutropenic patients with PA BSI (the IRONIC cohort).

## 2. Material and Methods

### 2.1. Study Design and Setting

This study is part of the IRONIC project: a large multicenter, international, retrospective cohort study conducted from 1 January 2006 to 31 May 2018 at 34 centers in 12 countries. The number of participating centers and the number of patients recruited at each one has previously been published (IRONIC) [13].

### 2.2. Participants

All adult (≥18 years) onco-hematological neutropenic patients including hematopoietic stem cell transplant (HSCT) recipients were eligible for the study if they were diagnosed with at least one episode of PA BSI during the study period. Subsequent episodes caused by PA occurring in the same patient were included in the study if they occurred at intervals of more than one month. The follow-up period was 30 days from BSI onset.

### 2.3. Variables

Data regarding the baseline characteristics, clinical and microbiological features, and the endpoints were collected. Empirical antibiotic therapy was considered when the antibiotic was administered before the reception of definitive susceptibility results. Adequate initial empirical antibiotic therapy was defined when patients received at least one in vitro active antibiotic against the PA strain. Initial IEAT was considered when the patient did not receive any empirical antibiotic with in vitro activity, or an empirical antibiotic therapy was lacking. The antipseudomonal β-lactams were uniformly administered at the current standard doses for the treatment of febrile neutropenia [14,15]. In the case of renal impairment, the dosing was adjusted accordingly.

### 2.4. Outcomes

Episodes of PA BSI occurring in HM patients were compared to those developing in patients with ST. Risk factors associated with 30-day mortality were investigated in both groups.

### 2.5. Microbiological Studies

Clinical samples were processed at the microbiology laboratories of each participating center in accordance with the standard operating procedures. PA was identified using standard microbiological techniques at each center. In vitro susceptibility was determined according to the EUCAST recommendations [19], except at a center in Lebanon and at one center in Argentina, where the CLSI breakpoints were used, and at the center in the UK where the BSAC recommendations were used before 2016 [20]. PA isolate phenotypes were stratified in accordance with recent standard definitions; multidrug-resistant (MDR) was defined as acquired non-susceptibility to at least one agent in three or more antimicrobial categories; extensively drug-resistant (XDR) was defined as non-susceptibility to at least one agent in all but two or fewer antimicrobial categories, as previously described [21].

### 2.6. Definitions

Neutropenia and profound neutropenia were defined as an absolute neutrophil count below 0.5 × 10^9^ cells/mm and 0.1 × 10^9^ cells/mm, respectively [14]. The Multinational Association for Supportive Care in Cancer (MASCC) score was calculated as described elsewhere [22]. Previous corticosteroid treatment was defined as the administration of ≥20 mg of prednisone, or equivalent dosing, for at least four weeks within 30 days of BSI onset. Bacteremic PA pneumonia was defined as the presence of an acute respiratory illness and a new pulmonary infiltrate on a chest radiograph and/or chest tomography in association with concurrent PA BSI.

Other BSI sources were established using standard U.S. Centers for Disease Control and Prevention criteria for secondary BSI [23]. In addition, the source of BSI was defined as unknown or endogenous in patients in whom no other sources were identified. Septic shock was defined as a systolic blood pressure <90 mmHg that was unresponsive to fluid treatment or required vasoactive drug therapy [24]. Mucositis was considered in patients with ulcerative lesions involving only the oral cavity. Comorbidities were defined as the presence of one or more of the following diseases: chronic obstructive pulmonary disease (COPD), chronic heart disease, chronic hepatic disease, diabetes mellitus, chronic renal disease, and cerebrovascular disease.

Persistent BSI was considered if the blood cultures were positive after 48 h of adequate antibiotic therapy. The 7-day and 30-day case-fatality rates were defined as death from any cause within 7 days and 30 days of BSI onset, respectively.

### 2.7. Statistical Analysis

To define the cohort characteristics, categorical variables were presented as the number of cases and percentages, while continuous variables were presented as the mean and standard deviation (SD) or median and interquartile range (IQR). Continuous variables were compared using the Student’s *t*-test or the Mann–Whitney U test where appropriate. Fisher’s exact test or Pearson’s χ2 test were applied to assess the relationship between categorical variables. Odds ratios (ORs) and 95% confidence intervals (CIs) were calculated. A *p* value of <0.05 was considered statistically significant. The analysis was performed with the stepwise logistic-regression model of the SPSS software package version 19.0 (SPSS Inc., Chicago, IL, USA).

### 2.8. Ethics

The study was approved by the Institutional Review Board at Bellvitge University Hospital (local reference number PR408/17) and by the local Research Ethics Committees at the participating centers. It was conducted in accordance with the guidelines of the Declaration of Helsinki. The need for informed consent was waived by the Clinical Research Ethics Committee due to the retrospective design.

## 3. Results

### 3.1. Clinical Characteristics

Of the 1217 episodes of PA BSI, 917 occurred in patients with HM and 300 in patients with ST. In hematological patients, the most frequent malignancies were acute leukemia 312 (34%), non-Hodgkin lymphoma 271 (29.6%), and acute lymphoblastic leukemia 97 (10.6%), and among them, 289 (31.5%) received a hematopoietic stem cell transplant. The most common malignancies among patients with solid tumors were lung cancer 89 (29.7%), lower gastrointestinal cancer 30 (10%), urinary cancer 29 (9.7%), breast cancer 28 (9.3%) and head and neck cancer 26 (8.7%). Baseline characteristics of the patients included in the study are detailed in Table 1. Patients with HM were significantly younger, had more frequent profound neutropenia, a higher MASCC index score and severe mucositis. They also more often received previous biological therapies, corticosteroids, and antibiotics. Patients with ST were more likely to have comorbidities than patients with HM, especially COPD, and had received more frequent previous chemotherapy. Catheter-related infection, mucositis, and perineal infection were more common in patients with HM, whereas pneumonia and other abdominal sources were more frequent in ST patients.

### 3.2. Etiology and Antibiotic Resistance

Hematological patients frequently had more BSI due to MDRPA (23.2% vs. 7.7%; *p* < 0.001) and XDRPA (17.2% vs. 5.3%; *p* < 0.001). The rate of polymicrobial infection did not differ between groups (17.9% vs. 17.7%).

### 3.3. Empirical Antibiotic Therapy and Clinical Outcomes

Empirical antibiotic therapy and clinical outcomes are described in Table 2. Patients with ST frequently received more monotherapy (73.6% vs. 57.5%; *p* < 0.001) whereas patients with HM were more likely to receive IEAT (20.1% vs. 12%; *p* < 0.001). Patients with ST presented more frequently with septic shock (45.8% vs. 30%; *p* < 0.001), and also presented worse outcomes, with increased 7-day (38% vs. 24.2%; *p* < 0.001) and 30-day (49% vs. 37.3%; *p* < 0.001) case-fatality rates.

### 3.4. Risk Factors for Overall Case-Fatality Rate

Risk factors for 30-day mortality are described in Table 3 and Table 4. High risk MASCC index score, IEAT, pneumonia, infection due to MDRPA and septic shock were associated with 30-day case fatality rate in hematologic patients (Table 3). A high risk MASCC index score, IEAT, persistent BSI, and septic shock were associated with the 30-day case fatality rate in patients with ST (Table 4). Therapy with the granulocyte colony-stimulating factor was associated with survival in both groups.

### 3.5. Discussion

This large, multicenter, international cohort study of high-risk neutropenic cancer patients identified several differences between hematological patients and those with solid tumors with PA BSI that can be useful to optimize their therapeutic management. Hematological patients frequently had more profound neutropenia and BSI from an endogenous source and from the catheter, and were more likely to receive IEAT due to higher rates of multidrug resistance among the PA isolates. Conversely, patients with solid tumors frequently had more COPD and pneumonia, and presented poorer outcomes, with higher rates of septic shock at presentation and higher case-fatality rates.

Of note, pneumonia was the second cause of PA BSI in both groups of patients, but it was significantly more frequent in patients with solid tumors. This is probably due to the fact that the latter group of patients often had COPD as a relevant comorbidity and lung cancer as the underlying disease. The presence of dysfunctional malignant cells in the lung tissue predisposes to invasive disease in these patients, who are often colonized by *Pseudomonas aeruginosa* [25]. In contrast, PA BSI in hematological patients was mainly associated with the administration of myeloablative chemotherapy and its consequences such as the presence of profound neutropenia and mucositis and the infection of long-term central venous catheters. Even though Gram-positives still remain the leading cause of catheter-related BSI, an increase in Gram-negatives has been described in the last decades [26,27,28].

An important finding of our study is that hematological patients frequently had more infections due to MDRPA and XDRPA and were more likely to receive IEAT. This is probably a consequence of the increased rates of previous antibiotic exposure in this group of patients as well as the use of quinolone prophylaxis [1,29,30]. Even though the reduction of antibiotic consumption is one of the major cornerstones in the fight against antibacterial resistance, this strategy may be difficult to accomplish in these high-risk patients. Nevertheless, in light of numerous studies that have identified the use of quinolone prophylaxis as a major risk factor for multidrug resistance, the universal use of this preventive strategy is no longer recommended, particularly when the rates of quinolone resistance among Gram-negatives are high in some centers [13,31,32].

Importantly, carbapenems and piperacillin-tazobactam were the most common inadequate empirical agents used in both groups. This finding is of special concern in the current era of widespread antimicrobial resistance and emerging resistance to carbapenems. Thus, the administration of combined empirical therapy and the prompt use of the recently available antibiotics in febrile cancer patients should be seriously considered [33,34,35,36,37].

As observed in other series of hematological patients with PA BSI, the early and overall case-fatality rates of the whole cohort were high [6,8,9,10,11,12]. Of note, outcomes were better in patients with hematological malignancies in spite of more frequently presenting persistent BSI and receiving IEAT more often. One plausible explanation is that solid tumor patients were older, with more comorbidities, more frequent advanced neoplasm, more septic shock at presentation, and pneumonia as the source of BSI.

Finally, a high MASCC index score, septic shock, and IEAT were all identified as risk factors for 30-day case fatality rates in both groups, whereas receiving therapy with granulocyte colony-stimulating factor (G-CSF) was associated with improved survival. It is noteworthy that the improvement in the empirical treatment and the administration of G-CSF were the only modifiable factors associated with mortality in our study.

The main strength of this study is that it is based on one of the largest cohorts of neutropenic cancer patients with PA BSI, with a multicenter international design that allows for generalization of the results. Nevertheless, this study also has some limitations that should be acknowledged. First, this is a retrospective study, so the main limitation of the data is related to the potential effects of unmeasured variables and residual confounding. Second, this was not a randomized clinical trial; thus, the choice of therapy may have been influenced by patient-related variables and by the clinical presentation. Finally, it was a relatively long period, and it is not possible to rule out a certain calendar effect in some variables such as mortality.

In conclusion, we identified significant differences between patients with hematological malignancies and solid tumors with PA BSI that should be considered when approaching cancer patients with suspected PA infection. Among the risk factors associated with overall mortality, IEAT and the administration of G-CSF were the only modifiable variables.

## Figures and Tables

**Table 1 pathogens-11-01132-t001:** The epidemiological and clinical characteristics of episodes of *Pseudomonas aeruginosa* bloodstream infection in patients with hematological malignancies and solid tumors.

Characteristics	Overall n = 1217	Hematological Malignancies n = 917 (%)	Solid Tumors n = 300 (%)	*p*-Value
Age (years, median, range)	60 (IQR 20)	59 (IQR 21)	64 (IQR 17)	<0.001
Male sex	751	559 (61.0)	192 (64.0)	0.192
Comorbidities	586	426 (48.4)	160 (57.1)	0.007
Chronic heart disease	149	120 (13.1)	29 (9.7)	0.069
Chronic obstructive pulmonary disease	100	54 (5.9)	46 (15.3)	<0.001
Diabetes mellitus	86	60 (6.5)	26 (8.7)	0.144
Chronic liver disease	68	52 (5.7)	16 (5.3)	0.478
Other comorbidities *	183	140 (15.3)	43 (14.3)	0.386
High risk MASCC * index score	341	269 (32.2)	72 (26.7)	0.050
Profound neutropenia (0.1 × 10^9^ cells/mm)	728	600 (67.0)	128 (44.6)	<0.001
Previous chemotherapy (1 month)	1037	764 (83.6)	273 (91.3)	<0.001
Previous biological therapies (3 months)	186	172 (19.2)	14 (4.8)	<0.001
Previous hospital admission (3 months)	744	574 (63.1)	170 (57.8)	0.062
Severe mucositis (grade III-IV)	169	142 (15.7)	27 (9.1)	0.002
Corticosteroid therapy (1 month)	832	727 (58.3)	105 (36.4)	<0.001
Previous antibiotic therapy (1 month)	665	591 (65.4)	74 (25.3)	<0.001
Prior quinolone prophylaxis	195	186 (20.5)	9 (3.0)	<0.001
Intravenous vascular catheter	908	756 (82.5)	152 (50.7)	<0.001
Urinary catheter	206	162 (18.1)	44 (15.1)	0.132
Nosocomial acquisition	694	620 (67.6)	74 (24.7)	<0.001
Source of bloodstream infection				
Endogenous source	751	348 (37.9)	107 (35.7)	0.261
Pneumonia	586	210 (22.9)	101 (33.7)	<0.001
Urinary tract	311	34 (3.7)	17 (5.7)	0.099
Catheter-related infection	112	98 (10.7)	14 (4.7)	0.001
Other abdominal *	112	26 (2.8)	19 (6.3)	0.006
Neutropenic enterocolitis	71	57 (6.2)	14 (4.7)	0.199
Mucositis	71	22 (2.4)	2 (0.7)	0.042
Skin and soft tissue infection	70	57 (6.2)	13 (4.3)	0.141
Unknown origin	70	13 (1.4)	2 (0.7)	0.244
Perineal infection	51	3 (3.6)	1 (0.3)	0.001
Septic shock at presentation	411	274 (30.0)	137 (45.8)	<0.001
Septic metastases	88	78 (8.9)	10 (3.4)	<0.001
Gangrenous ecthyma	51	50 (5.5)	1 (0.3)	<0.001

* Other comorbidities included chronic renal disease and cerebrovascular disease. MASCC (Multinational Association for Supportive Care in Cancer). Other abdominal sources included cholangitis, peritonitis, and intraabdominal abscesses.

**Table 2 pathogens-11-01132-t002:** The empirical antibiotic therapy and clinical outcomes of *Pseudomonas aeruginosa* bloodstream infection compared by groups.

Characteristics	Hematological Malignancies n = 917 (%)	Solid Tumors n = 300 (%)	*p*-Value
**Empirical Antibiotic Therapy**			
Monotherapy	519 (57.5)	218 (73.6)	0.000
Combination therapy	383 (42.5)	78 (26.4)	<0.001
β-lactam + aminoglycoside	209 (23.2)	51 (17.2)	0.031
β-lactam + non-aminoglycoside	55 (6.1)	17 (5.7)	0.821
Initial inadequate empirical antibiotic therapy	184 (20.1)	36 (12.0)	<0.001
Carbapenems	89	10	
Piperacillin-tazobactam	52	13	
Glycopeptides	35	7	
Aminoglycosides	18	1	
Broad-spectrum cephalosporins	15	1	
Quinolones	11	1	
Other	42	17	
Granulocyte colony-stimulating factor	490 (54)	136 (45.9)	0.009
**Clinical Outcomes**			
Intensive care unit admission	285 (31.1)	103 (34.3)	0.164
Invasive mechanical ventilation	187 (20.4)	59 (19.7)	0.422
Persistent bloodstream infection	111 (12.3)	23 (7.9)	0.024
Early case-fatality rate (7 days)	222 (24.2)	114 (38.0)	<0.001
Overall case-fatality rate (30 days)	342 (37.3)	147 (49.0)	<0.001

**Table 3 pathogens-11-01132-t003:** The risk factors for the overall case-fatality rate in hematological patients by univariate and multivariate analysis.

Characteristics	n	Adjusted OR (95% CI)	*p*-Value
Age		1.01 (0.99–1.02)	0.088
Male sex	559 (61.0)	0.74 (0.51–1.08)	0.126
High risk MASCC index score	269 (32.2)	2.53 (1.63–3.95)	<0.001
Initial inadequate empirical antibiotic therapy	184 (20.1)	3.45 (1.09–10.92)	<0.035
Persistent bloodstream infection	111 (12.3)	1.43 (0.83–2.47)	0.195
Granulocyte colony-stimulating factor	490 (54)	0.53 (0.36–0.76)	0.001
Pneumonia	210 (22.9)	2.23 (1.47–3.38)	<0.001
MDRPA	157 (17.2)	2.22 (1.44–3.41)	<0.001
Septic shock	274 (30.0)	7.12 (4.78–10.60)	<0.001

MDRPA: Multidrug-resistant *Pseudomonas aeruginosa.*

**Table 4 pathogens-11-01132-t004:** The risk factors for the overall case-fatality rate in patients with solid tumors by univariate and multivariate analysis.

Characteristics	n	Adjusted OR (95% CI)	*p*-Value
Age		1.01 (0.98–1.03)	0.329
Male sex	192 (64.0)	0.98 (0.52–1.83)	0.949
High risk MASCC index score	72 (26.7)	2.68 (1.21–5.94)	0.015
Inadequate initial empirical antibiotic therapy	36 (12.0)	2.84 (1.10–7.35)	0.031
Persistent bloodstream infection	23 (7.9)	9.92 (2.08–47.20)	0.004
Granulocyte colony-stimulating factor	136 (45.9)	0.26 (0.14–0.48)	<0.001
Septic shock	137 (45.8)	3.97 (2.10–7.51)	<0.001

## Data Availability

The datasets generated during and/or analysed during the current study are available from the corresponding author on reasonable request.

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
