# Peer review of "Pseudomonas aeruginosa Bloodstream Infections in Patients with Cancer: Differences between Patients with Hematological Malignancies and Solid Tumors"

_pathogens, 2022, doi:10.3390/pathogens11101132_

Round 1

Reviewer 1 Report

Congrats to the authors for completing this work! 

Here are my review comments to improve the manuscript draft further -

1. IEAT is referred to as Inappropriate EAT in some places, and Inadequate EAT in others. Suggest using the first one uniformly throughout the draft.

2. Expand IRONIC study - to differentiate it from IRONIC trial

3. Expand abbreviations at their first mention in the draft

4. What do other abdominal sites refer to?

5. Use either PABSI or PA BSI throughout

6. Use either British or US English throughout. At present, they are both used - E.g., tumor and tumour, hematology

7. The discussion about antibiotic exposure in patients with hematological malignancies needs further work - does it mean that previous antibiotic exposure leads to more resistant organisms being isolated? And yet, more sepsis was observed in ST. Any reasons for this you may want to put forth? It will be useful.

8. Is there a link for the web-based calculator?

9. Suggest inserting references for the statement - "The presence of dysfunctional malignant cells in the lung tissue predisposes to invasive disease in these patients who are often colonized by Pseudomonas aeruginosa."

10. Check references and their formatting, especially reference 20

11. Some editorial corrections will include deletion of extra spaces, punctuations, including space between number and unit, etc. Have highlighted them in the pdf, including in Table 1.

Best regards,

Author Response

Reviewer #1

Congrats to the authors for completing this work! 

Here are my review comments to improve the manuscript draft further.

Authors: Thank you for your positive feedback on the article, we have included your comments in the original draft marked with "track changes" and below we detail our responses.

Question #1. IEAT is referred to as Inappropriate EAT in some places, and Inadequate EAT in others. Suggest using the first one uniformly throughout the draft. 

Authors: Thanks for the comment, we have unified the term as Inadequate EAT (IAET) throughout the manuscript.

Question #2. Expand IRONIC study - to differentiate it from IRONIC trial

Authors: The aim of the first study obtained from the IRONIC database was to assess the prevalence and evolution of multidrug resistance among the P. aeruginosa isolates causing bacteremia in neutropenic onco-hematological patients during the study period (2006-2018), and also to build a clinical prediction model to predict multidrug resistance. Other secondary outcomes included the early and overall 30-day mortality rates and the rate of inadequate initial empirical antibiotic therapy, comparing the episodes caused by multidrug-resistant isolates with those caused by susceptible strains. The whole cohort included onco-hematological patients together. The aim of the current study is to identify the differences on clinical features, antibiotic therapy and outcomes according to the specific underlying condition (hematological malignancies versus solid tumors).

Question #3. Expand abbreviations at their first mention in the draft.

Authors: According to the reviewer’s suggestion, we have expanded the abbreviations at their first mention in the draft.

Question #4. What do other abdominal sites refer to?

Authors: Other abdominal sites refer to abdominal infections that do not include endogenous bacteremia, neutropenic enterocolitis, and perianal infection, such as cholangitis, peritonitis, and intraabdominal abscesses. We have included the definition in the foot of Table 1.

Question #5. Use either PABSI or PA BSI throughout

Authors: Thank you, we have unified the term as PA BSI throughout the manuscript.

Question #6. Use either British or US English throughout. At present, they are both used - E.g., tumor and tumour, hematology.

Authors: Thank you. We have unified the spelling to American English throughout the study.

Question #7. The discussion about antibiotic exposure in patients with hematological malignancies needs further work - does it mean that previous antibiotic exposure leads to more resistant organisms being isolated? And yet, more sepsis was observed in ST. Any reasons for this you may want to put forth? It will be useful.

Authors: Thank you for your comment. Previous antibiotic exposure has been widely recognized as a risk factor for multidrug resistance in the literature, this is way we have not discussed it further. Nevertheless, we have added some references in order to support the evidence (see line 419, References #1, #29 and #30).

Regarding sepsis in ST patients, a possible explanation is that patients with solid tumors had more frequently pneumonia compared with patients with hematological patients, and bacteremic pneumonia is a severe clinical manifestation that often presents with septic shock.

Question #8. Is there a link for the web-based calculator?

Authors: Yes, there is a link for the web-based calculator which was provided in the first manuscript of the IRONIC project (http://ubidi.shinyapps.io/ironic). We have included it in the text (line 187).

Question #9. Suggest inserting references for the statement - "The presence of dysfunctional malignant cells in the lung tissue predisposes to invasive disease in these patients who are often colonized by Pseudomonas aeruginosa."

Authors: According to the reviewer’s suggestion, we have inserted a reference regarding this issue (reference #25 by Lanoix et al., line 333)

Question #10. Check references and their formatting, especially reference 20.

Authors: Thank you, we have checked all the references throughout the manuscript.

Question #11. Some editorial corrections will include deletion of extra spaces, punctuations, including space between number and unit, etc. Have highlighted them in the pdf, including in Table 1.

Authors: Thank you for the editorial corrections, to which we agree.

Reviewer 2 Report

The cases in solid group is 3 times lower vs hematologic 

HM. I think it should be noted that explored how the differences in numbers affect results and conclusions 

Author Response

Reviewer #2

Question #1: The cases in solid group is 3 times lower vs hematologic. I think it should be noted that explored how the differences in numbers affect results and conclusions 

Authors: We agree with the reviewer that the number of episodes of PA BSI was three times higher in hematological patients than in patients with solid tumors. These is due to the fact that chemotherapy-induced profound and prolonged neutropenia is more common in hematological patients, and consequently, BSI occurs also more frequently in these patients, including PA BSI. For this reason, we decided to evaluate the two groups separately, not only in the descriptive analysis, but also in the analysis of risk factors for mortality.  

Reviewer 3 Report

Pseudomonas aeruginosa bloodstream infections in patients with cancer: differences between patients with haematological malignancies and solid tumours, C. Royo-Cebrecos et al.

This is a large multicentre, retrospective cohort study. Here the authors present results on patients which fit the eligibility criteria from 34 different centres from 12 countries over a twelve-year period. The focus of this work included rates of multidrug resistance, source of infection and risk factors for 30-day mortality Pseudomonas aeruginosa bloodstream infections comparing patients with haematological malignancies to those with solid tumours.

There is a considerable amount of data presented succinctly here. This is a well written paper which presents the data on what they set out to. I have several minor suggestions outlined below to help clarify definitions throughout the paper and maintain consistency for the reader. Overall, I think more editing/proofreading is needed to improve the manuscript.

Title: In the title haematological and tumour are spelt using the British English spelling while the rest of the manuscript uses hematological and tumor the American English spelling. Can you be consistent throughout with which spelling you chose?

Line 130: Please define MASCC

Line 146: Please define G-CSF

PABSI: Sometimes there is a space between the A and the B (i.e., PA BSI) and other times there is not (i.e., PABSI). Please be consistent throughout the manuscript

2.5 Microbiological studies: Please add in a definition for MDRPA and XDRPA. I am not sure which section in the methods is most appropriate, but these should be described.

Line 223: Severe neutropenia is defined, but elsewhere in the manuscript the term profound neutropenia is used. According to the definition, I believe all references to profound neutropenia (Lines 129, 262, 304, 317, Table 1) should be amended to match the definition and the word profound changed to severe.

Neutropenia: the definition varies throughout the manuscript.

Line 129 (Abstract): <100/mm3

Line 224: 0.1 x 109 cells/mm

Table 1: 0.1 109/litre

Can you please be consistent with how neutropenia is described throughout the manuscript.

Lines 235-238: Here a list of comorbidities is written out. When this is compared to what is listed in Table 1 there a few differences. Heart disease vs chronic heart disease, hepatic disease vs chronic liver disease and then in the definitions renal failure and cerebovasular disease are mentioned but in the table there is other comorbidities. Could you be more clear with the comorbidities and have them align in both sections.

3.1 Clinical characteristics: I understand that you describe the clinical cohort in more detail in C. Gudiol AAC 2020, but it would be beneficial to describe the two cohorts in more detail for this manuscript. Please include the breakdown of which haematological disease and which solid tumour are included in both cohorts. For example, HM may include leukemia, lymphoma and multiple myeloma while ST may include lung cancer and breast cancer. A list of the specific cancers would be helpful. I think this could be briefly described in 3.1 Clinical Characteristics but doesn’t need to go into Table 1.

Table 1: When I tallied up each of the individual comorbidities the total did not match what is published.

Overall comorbidity =586, but I counted 583

HM comorbidity = 426, but I counted 424

ST comorbidity = 160, but I counted 159

Can these figures be reviewed?

I found the definitions in 2.3 Variables and the results presented in 3.3 Empirical antibiotic therapy and clinical outcome and Table 2 every difficult to reconcile. Below are a few point I found confusing.

Firstly, there is five definitions described in 2.3

1.       Empirical antibiotics (AB) therapy: when AB are administered before susceptibility results are available

2.       Appropriate empirical AB therapy: when patients received at least one active AB against PA

3.       Empirical combination therapy: considered appropriate when both empirically administered AB are active against PA

4.       Appropriate monotherapy: when 2 AB are administered but only one is active against PA

5.       Inappropriate empirical AB therapy: when a patient does not receive any empirical AB with in vitro activity or when empirical AB therapy was lacking

Line 279: Patients with ST received more frequently initial empirical combination therapy (73.6% vs 57.7%). In Table 2 this result is next to the characteristic monotherapy. Should this be amended, monotherapy replaced with empirical combination therapy?

In Table 2, there are two characteristics described i) combination therapy empirically and ii) both AB with in vitro activity. Firstly, there is no definition for either characteristic defined in 2.3. Secondly, if combination therapy empirically is the same as the empirical combination therapy (as defined in 2.3), then isn’t this the same as both AB with in vitro activity and therefore shouldn’t these numbers be combined? If not, can you please use the definitions/wording as described in 2.3 for the characteristic headings in Table 2.

Of the five antibiotic definitions (2.3) only 2, IEAT and combination, are referred to in the results and discussion. If there are results for the other three definitions and you believe them to be of value to this manuscript, can you please include. Conversely, if these results are included in Table 2 but the characteristic titles, as they currently stand, are not in line with the definitions can you please fix. Clarification and consistency with these definitions are important. However, if results for these other three definitions are not included in the manuscript, then are the definitions needed? They add to confusion if they are not referenced in the body of the manuscript.

Table 2: There is a list of antibiotics, carbapenems, piperacillin-tazobactam, glycopeptides….etc. What are these listed in relation to? Are these monotherapies which the patients were administered? Maybe a subheading above the list of antibiotics will make this clearer.

Table 2: Towards the bottom of Table 2 there is a list of clinical outcomes, a heading, such as clinical outcome might be helpful here to highlight that these are not part of the patient treatment but rather health outcomes.

Line 135, Line 204, Line 280, Table 2: Please be consistent with your definition and description of IEAT. Should it be inadequate initial AB therapy (Line 135, Line 280, Table 2) or inappropriate empirical AB treatment (Line 204)? Please amend throughout.

Line 273 & 274: When the results are the reported for MDRPA & XDRPA for BSI are these for patient with a solely with PA?

Line 274: Should extensively-resistant PA be changed to extensively drug resistant PA (XDRPA) as referred to on Line 322?

Line 274: Here it mentions that haematological patients have more BSI due to XDRPA with the numbers 17.2% vs 5.3% however in the abstract (Lines 134/135) this exact figure is said to be due to MDRPA not XDRPA. Can you please fix so that both statements reflect the same result?

Line 274 & 275: Polymicrobial infections – does this include PA and another bacteria or two or more bacteria (not PA)?

Line 290: Where infection due to MDRPA is mentioned is this all PA infections or just BSI?

Line 292: Where persistent BSI is mentioned, does this refer to PA BSI only or is this any BSI, including mono & poly microbial? Persistent BSI is also mentioned in the tables (3&4).

Table 4: MDRPA is not listed as a characteristic. I assume it is not significant as it is not listed but it might be helpful to see this data alongside of the HM patients as a comparison.

Reporting of Gender throughout the manuscript: In Table 1 this is reported as Male sex, Table 3 as sex and Table 4 as sex male, could you please be consistent throughout?

Use of the word comorbidities: This is not consistent throughout the manuscript. It switches between comorbidities and co-morbidities. Please be consistent

Author Response

Reviewer #3

Pseudomonas aeruginosa bloodstream infections in patients with cancer: differences between patients with haematological malignancies and solid tumours, C. Royo-Cebrecos et al.

This is a large multicentre, retrospective cohort study. Here the authors present results on patients which fit the eligibility criteria from 34 different centres from 12 countries over a twelve-year period. The focus of this work included rates of multidrug resistance, source of infection and risk factors for 30-day mortality Pseudomonas aeruginosa bloodstream infections comparing patients with haematological malignancies to those with solid tumours.

There is a considerable amount of data presented succinctly here. This is a well written paper which presents the data on what they set out to. I have several minor suggestions outlined below to help clarify definitions throughout the paper and maintain consistency for the reader. Overall, I think more editing/proofreading is needed to improve the manuscript.

Authors:Thank you for your positive feedback on the article, we have included your comments in the original draft marked with "track changes", and below we detail our responses.

Question #1Title: In the title haematological and tumour are spelt using the British English spelling while the rest of the manuscript uses hematological and tumor the American English spelling. Can you be consistent throughout with which spelling you chose?

Authors: Please see answer to question #6 of Reviewer #1.

Question #2. Line 130: Please define MASCC

Authors: We have added the definition of MASCC (line 134 and 256).

Question #3. Line 146: Please define G-CSF

Authors: We have added the definition of G-CSF (line 440).

Question #4. PABSI: Sometimes there is a space between the A and the B (i.e., PA BSI) and other times there is not (i.e., PABSI). Please be consistent throughout the manuscript

Authors: Please see answer to question #5 of Reviewer #1.

Question #5. 2.5 Microbiological studies: Please add in a definition for MDRPA and XDRPA. I am not sure which section in the methods is most appropriate, but these should be described.

Authors: Thank you for the comment. We have included the definitions following Magiorakos AP et al (lines 248-252).

Question #6. Line 223: Severe neutropenia is defined, but elsewhere in the manuscript the term profound neutropenia is used. According to the definition, I believe all references to profound neutropenia (Lines 129, 262, 304, 317, Table 1) should be amended to match the definition and the word profound changed to severe.

Authors: Thank you for your comment. The widely recognized definition of neutropenia is an absolute neutrophil count (ANC) 0.5×109 cells/mm, whereas profound neutropenia refers to an ANC <0.1×109 cells/100/mm. We have kept the term “profound neutropenia” instead of “sever neutropenia” because it is the term used in the clinical practice and in the guidelines. We have added the reference #14 (line 255).

Question #7. Neutropenia: the definition varies throughout the manuscript.

Line 129 (Abstract): <100/mm3

Line 224: 0.1 x 109 cells/mm

Table 1: 0.1 109/litre

Can you please be consistent with how neutropenia is described throughout the manuscript.

Authors: Thank you. We have unified the definition of neutropenia as 0.1×109 cells/mm throughout the manuscript (see lines 133, 255) and Table 1.

Question #8. Lines 235-238: Here a list of comorbidities is written out. When this is compared to what is listed in Table 1 there a few differences. Heart disease vs chronic heart disease, hepatic disease vs chronic liver disease and then in the definitions renal failure and cerebovasular disease are mentioned but in the table there is other comorbidities. Could you be more clear with the

comorbidities and have them align in both sections.

Authors: Thank you for the comment. We have unified the criteria and we have specified “other co-morbidities” in the footnote of Table 1.

Question #9. 3.1 Clinical characteristics: I understand that you describe the clinical cohort in more detail in C. Gudiol AAC 2020, but it would be beneficial to describe the two cohorts in more detail for this manuscript. Please include the breakdown of which haematological disease and which solid tumour are included in both cohorts. For example, HM may include leukemia, lymphoma and multiple myeloma while ST may include lung cancer and breast cancer. A list of the specific cancers would be helpful. I think this could be briefly described in 3.1 Clinical Characteristics but doesn’t need to go into Table 1.

Authors: Following the reviewer’s suggestion, we have included more detailed information regarding the characteristics of each group of patients (lines 314-319).

Question #10. Table 1: When I tallied up each of the individual comorbidities the total did not match what is published.

Overall comorbidity =586, but I counted 583

HM comorbidity = 426, but I counted 424

ST comorbidity = 160, but I counted 159

Can these figures be reviewed?

Authors: Thank you. We have modified the co-morbidities in Table 1.

Question #11. I found the definitions in 2.3 Variables and the results presented in 3.3 Empirical antibiotic therapy and clinical outcome and Table 2 every difficult to reconcile. Below are a few point I found confusing.

Authors: Thank you. We agree with the reviewer that there are too many definitions provided for the empirical antibiotic therapy, and this is confusing. Therefore, we have decided to simplify the definitions and keep only the most important definitions (empirical antibiotic, adequate empirical antibiotic and inadequate empirical antibiotic), and remove the others because they do not provide any valuable information. We have modified Table 2 accordingly. (see lines 230-234 and Table 2).

Question #12. Line 279: Patients with ST received more frequently initial empirical combination therapy (73.6% vs 57.7%). In Table 2 this result is next to the characteristic monotherapy. Should this be amended, monotherapy replaced with empirical combination therapy?

Authors: Thank you, we have modified the error type (line 361).

Question #13. In Table 2, there are two characteristics described i) combination therapy empirically and ii) both AB with in vitro activity. Firstly, there is no definition for either characteristic defined in 2.3. Secondly, if combination therapy empirically is the same as the empirical combination therapy (as defined in 2.3), then isn’t this the same as both AB with in vitro activity and therefore shouldn’t these numbers be combined? If not, can you please use the definitions/wording as described in 2.3 for the characteristic headings in Table 2.

Authors: Thank you, please see answer to question #11.

Question #14. Of the five antibiotic definitions (2.3) only 2, IEAT and combination, are referred to in the results and discussion. If there are results for the other three definitions and you believe them to be of value to this manuscript, can you please include. Conversely, if these results are included in Table 2 but the characteristic titles, as they currently stand, are not in line with the definitions can you please fix. Clarification and consistency with these definitions are important. However, if results for these other three definitions are not included in the manuscript, then are the definitions needed? They add to confusion if they are not referenced in the body of the manuscript.

Authors: Thank you, please see answer to question #11.

Question #15. Table 2: There is a list of antibiotics, carbapenems, piperacillin-tazobactam, glycopeptides….etc. What are these listed in relation to? Are these monotherapies which the patients were administered? Maybe a subheading above the list of antibiotics will make this clearer.

Authors: This list of antibiotics corresponds to initial inadequate empirical antibiotic therapy. We have marked it in bold letters to identify it as a subheading.

Question #16. Table 2: Towards the bottom of Table 2 there is a list of clinical outcomes, a heading, such as clinical outcome might be helpful here to highlight that these are not part of the patient treatment but rather health outcomes.

Authors: Thank you. We have added a subheading in the table.

Question #17. Line 135, Line 204, Line 280, Table 2: Please be consistent with your definition and description of IEAT. Should it be inadequate initial AB therapy (Line 135, Line 280, Table 2) or inappropriate empirical AB treatment (Line 204)? Please amend throughout.

Authors: Thank you. We have unified the term IEAT to inadequate empirical antibiotic therapy throughout the manuscript.

Question #18. Line 273 & 274: When the results are the reported for MDRPA & XDRPA for BSI are these for patient with a solely with PA?

Authors: No, they aren’t, but the great majority of episodes due to MDRPA and XDRPA were monomicrobial.

Question #19. Line 274: Should extensively-resistant PA be changed to extensively drug resistant PA (XDRPA) as referred to on Line 322?

Authors: Indeed, we have changed the term (lines 344 and 414).

Question #20. Line 274: Here it mentions that haematological patients have more BSI due to XDRPA with the numbers 17.2% vs 5.3% however in the abstract (Lines 134/135) this exact figure is said to be due to MDRPA not XDRPA. Can you please fix so that both statements reflect the same result?

Authors: Thanks for the correction, we have changed the numbers in the abstract (line 139).

Question #21. Line 274 & 275: Polymicrobial infections – does this include PA and another bacteria or two or more bacteria (not PA)?

Authors: Polymicrobial infections include PA with another bacteria.

Question #22. Line 290: Where infection due to MDRPA is mentioned is this all PA infections or just BSI?

Authors: All infections in the manuscript refer to BSI.

Question #23. Line 292: Where persistent BSI is mentioned, does this refer to PA BSI only or is this any BSI, including mono & poly microbial? Persistent BSI is also mentioned in the tables (3&4).

Authors: Persistent BSI refers to all BSI episodes, including mono and polymicrobial episodes, but the definition only included the cases where PA persisted in the blood >48 hours after adequate therapy.

Question #24. Table 4: MDRPA is not listed as a characteristic. I assume it is not significant as it is not listed but it might be helpful to see this data alongside of the HM patients as a comparison.

Authors: Thank you for your comment. Indeed, we did not include MDRPA in the table because it did not reach statistical significance in the univariate analysis.

Question #25. Reporting of Gender throughout the manuscript: In Table 1 this is reported as Male sex, Table 3 as sex and Table 4 as sex male, could you please be consistent throughout?

Authors: Thank you, we have unified it as male sex in Tables 1, 3 and 4.

Question #26. Use of the word comorbidities: This is not consistent throughout the manuscript. It switches between comorbidities and co-morbidities. Please be consistent.

 Authors: Thank you for the comment, we changed it as comorbidities throughout the manuscript.
